# Interleaved High Voltage Gain DC-DC Converter with Winding-Cross-Coupled Inductors and Voltage Multiplier Cells for Photovoltaic Systems

**Shin-Ju Chen** [1,*] **, Sung-Pei Yang** [2] **, Chao-Ming Huang** [1] **, Sin-Da Li** [1] **and Cheng-Hsuan Chiu** [1]

[1] Department of Electrical Engineering, Kun-Shan University, Tainan 710303, Taiwan; cmhuang@mail.ksu.edu.tw (C.-M.H.); s104002713@g.ksu.edu.tw (S.-D.L.); s112001074@g.ksu.edu.tw (C.-H.C.)
[2] Department of Engineering Science, National Cheng Kung University, Tainan 701401, Taiwan; z11302001@ncku.edu.tw
* Correspondence: sjchen@mail.ksu.edu.tw

**Abstract:** An interleaved high voltage gain DC-DC converter with winding-cross-coupled inductors (WCCIs) and voltage multiplier cells is proposed for photovoltaic systems. The converter configuration is based on the interleaved boost converter integrating the diode-capacitor clamp circuits, the winding-cross-coupled inductors, and voltage multiplier cells to increase the voltage gain and reduce the semiconductor voltage stresses. The equal current sharing of two phases is achieved with the help of the winding-cross-coupled inductors. The converter achieves high voltage gain while operating at a proper duty ratio. The low-voltage-rated MOSFETs with low on-resistance are available to reduce the conduction losses due to the low switch voltage stress. The leakage energy of the coupled inductors is recycled such that the voltage spikes on the power switches are avoided. The input current ripple is decreased due to the interleaved operation. The operating principle and steady-state analysis of the proposed converter are proposed in detail. The design guidelines of the proposed converter are given. In addition, the closed-loop controlled system of the proposed converter is designed to diminish the effect of the variations in input voltage and load on the output voltage. Finally, the experimental results of a 1000 W converter prototype with 36 V input and 400 V output are given to validate the theoretical analysis and the converter performance.

**Keywords:** interleaved high voltage gain converter; winding-cross-coupled inductor; voltage multiplier cell

## 1. Introduction

Due to the global warming problem, the reduction in greenhouse gas emissions is one of the most significant methods. Renewable energy power systems have become increasingly important to achieve the goal of net zero emissions. Renewable energy sources such as photovoltaic (PV) and fuel cells often play a central role in distributed systems.

There are two kinds of PV grid-connected systems [1,2], as shown in Figure 1. The PV system with a high-voltage DC bus is shown in Figure 1a. Each PV module connects a high step-up DC-DC converter to the high-voltage DC bus and then it uses a single inverter to convert the DC power to the AC grid. Figure 1b shows a PV grid-connected system without a high-voltage DC bus. Each PV module is connected to the AC grid via a high step-up DC-DC converter and a DC-AC inverter. The output voltage of a PV module is generally 20–50 V and it cannot provide enough DC voltage for generating AC line voltage. If the AC grid voltage is 220 Vac, a 380/760 V DC bus voltage is required for the full-bridge/half-bridge inverter. Therefore, the high voltage gain DC-DC converters are needed in the PV systems to connect the PV modules with the high-voltage DC bus due to the low voltage generated by the PV modules. Then, an inverter is used to convert the voltage of the DC bus to the AC grid. In addition, the high voltage gain DC-DC converters

are also used in data centers and electric vehicles [3,4]. Consequently, the high voltage gain DC-DC converter is a topic worthy of study.

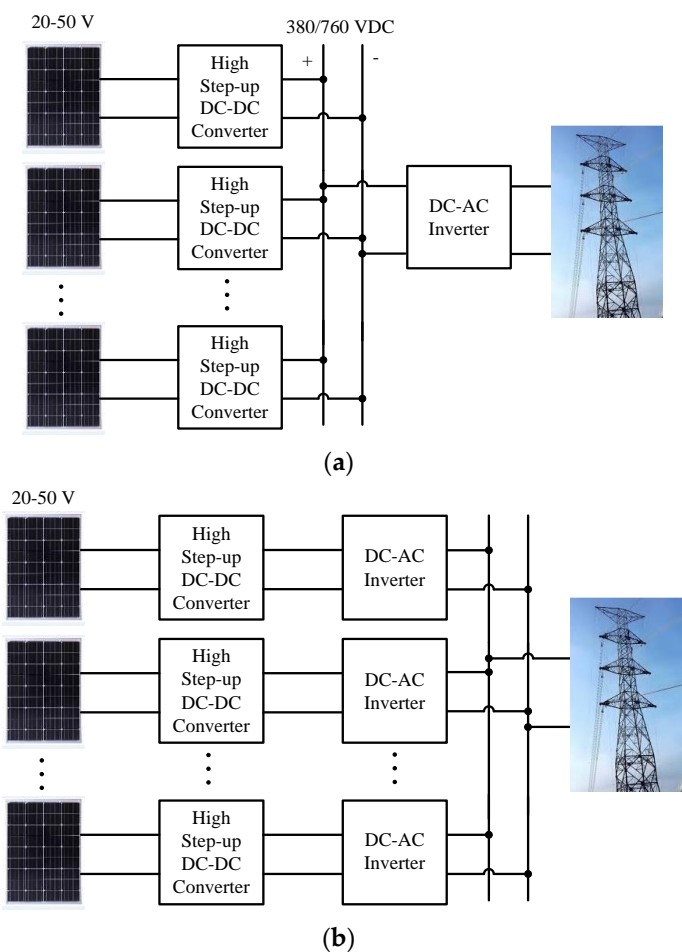

**Figure 1.** A PV grid-connected system. (**a**) With high voltage DC bus; (**b**) Without high voltage DC bus.

A conventional boost DC-DC converter can provide high voltage gain with an extreme duty ratio theoretically. However, the voltage gain is practically limited due to the parasitic effect. A boost DC-DC converter with extreme duty ratio operation will lead to a large current ripple, severe diode reverse-recovery problem, and high switching losses [5]. Furthermore, the high voltage stress on the switch and the diode results in large conduction losses and switching losses. These problems are the main limitations of conventional boost converters for high-voltage gain applications. The isolated converter topologies like the flyback DC-DC converter can achieve high voltage gain by selecting the high turns ratio of the transformer. However, the leakage inductance can cause high voltage spike such that a high-voltage-rated switch is needed. In order to overcome the limitations and problems, the high voltage gain DC-DC converters have become one of the research topics in the field of power electronics in the recent years. Many high voltage gain DC-DC converter topologies have been proposed in the literature.

The research review of the high step-up DC-DC converters and voltage-boost techniques are presented in [5–7], which are very worthy of reference. The coupled inductor technique is a common method for high voltage gain DC-DC converters [8–11]. The turns ratio of the coupled inductor can be used as a design freedom of the voltage gain. The switched inductor and switched capacitor technologies [12–14] are employed to achieve high voltage gain. It has the advantages of simple circuit configuration and low voltage stress on the switch and diode. The interleaved high step-up DC-DC converters with

voltage multiplier cells, composed of the coupled inductor, diode, and capacitor, are proposed in [15,16], which can achieve high voltage gain without operating at an extreme duty ratio. In order to reduce the switching losses, high step-up converters with zero-voltage switching performance are proposed to reduce the switching losses [17–20]; however, the converter has more power switches and the driving circuit is more complex. The winding-cross-coupled inductor technique for the high voltage gain converter has been proposed in [21–24]. The performance comparison is made in this article.

A novel interleaved high voltage gain DC-DC converter with winding-cross-coupled inductors and voltage multiplier cells is proposed in this article. The proposed converter is suitable for the requirement in the PV grid-connected systems. The features of the proposed converter are as follows:

(1) The high voltage gain can be achieved without working at an extreme duty ratio;
(2) The voltage stresses on the semiconductor devices are low such that the low-voltage-rated MOSFETs with low on-resistance $R_{ds(on)}$ and diodes with low forward voltage drop can be selected to reduce the conduction losses;
(3) The input current ripple is reduced by the interleaved operation;
(4) The diode reverse-recovery problem is alleviated due to the leakage inductances of the coupled inductors;
(5) The leakage energy of the coupled inductors is recycled such that the voltage spikes are avoided during the switch turned-off transient.

The proposed converter with these features is suitable for the applications of high voltage gain, high efficiency, and high power. A 1000 W laboratory prototype with 36 V input and 400 V output is implemented. The experimental results are provided to validate the performance of the proposed converter.

## 2. Converter Configuration and Operation Principles

The configuration of the proposed high voltage gain DC-DC converter with two phases is shown in Figure 2, where $S_1$ and $S_2$ are the power switches with the parasitic capacitors $C_{S1}$ and $C_{S2}$, respectively; $C_1$ and $C_2$ are the clamp capacitors; $C_3$ and $C_4$ are the switched capacitors; $C_5$ and $C_6$ are the voltage-doubler capacitors; $C_o$ is the output capacitor; $D_1$ and $D_2$ are the clamp diodes; $D_3$ and $D_4$ are the switched diodes; $D_5$ and $D_6$ are the voltage-doubler diodes; and $D_7$ and $D_8$ are the output diodes. There are two winding-cross-coupled inductors (WCCIs) in the proposed converter. The coupling reference of the WCCIs is denoted by "$*$" and "$\bullet$". The primary winding of each WCCI with $n_1$ turns is used as the filter inductor. The secondary winding with $n_2$ turns couples to the inductor in its phase and the tertiary winding with $n_3$ turns couples to the inductor in another phase. The voltage multiplier cell (VMC) consists of two diodes and two capacitors together with the secondary winding and the tertiary winding in series, which is used to increase the voltage gain and reduce the semiconductor voltage stresses.

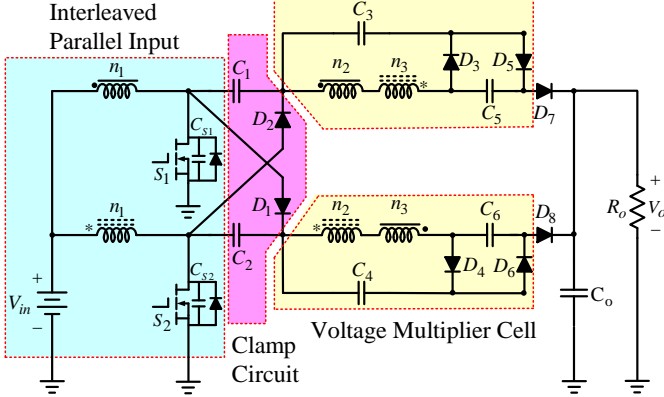

**Figure 2.** Proposed high voltage gain of the DC-DC converter.

The circuit model of the coupled inductor is presented as a combination of an ideal transformer, a magnetizing inductance, and two leakage inductances. $L_{m1}$ and $L_{m2}$ are the magnetizing inductances; $L_{k1}$ and $L_{k2}$ are the leakage inductances in the primary windings of WCCIs; $L_{s1}$ is the summation of the leakage inductances in the secondary winding of WCCI I and the tertiary winding of WCCI 2; and $L_{s2}$ is the summation of the leakage inductances in the tertiary winding of WCCI 1 and the secondary winding of WCCI 2. The equivalent circuit of the proposed converter is shown in Figure 3.

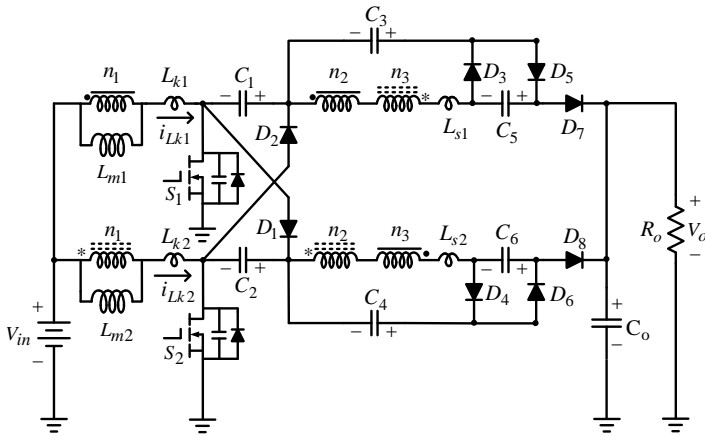

**Figure 3.** Equivalent circuit of the proposed converter.

The switches $S_1$ and $S_2$ operate in the interleaved mode with $180°$ phase shift and the same duty ratio. The duty ratio is greater than 0.5 to obtain high voltage gain. The parallel input configuration with interleaved operation reduces the input ripple current.

In this article, assuming that the coil turns $n_3$ is equal to $n_2$, the turns ratio is defined as $n = n_2/n_1 = n_3/n_1$ for the two WCCIs. When the proposed converter is operated in the continuous conduction mode (CCM), the steady-state waveforms of the proposed converter are shown in Figure 4.

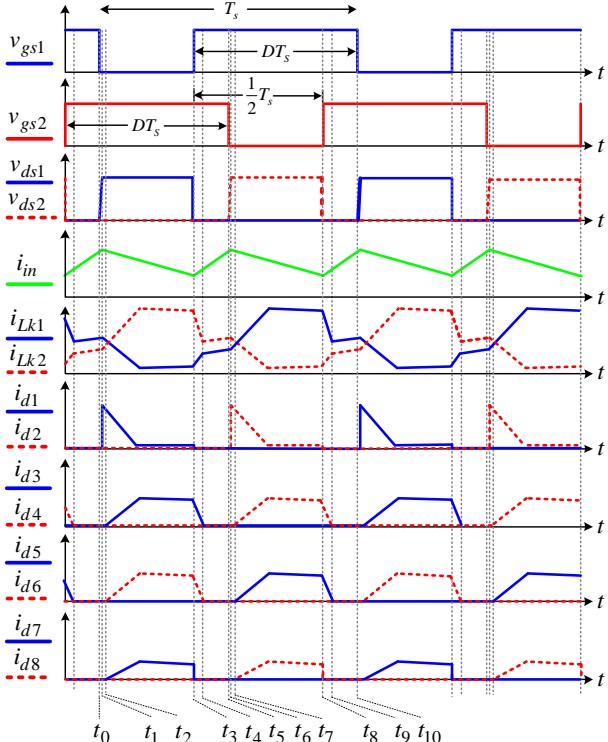

**Figure 4.** Steady-state waveforms.

There are 10 operational stages in one switching period and the equivalent circuits for each stage are shown in Figure 5. Due to the symmetrical structure, only five operational stages are chosen to analyze.

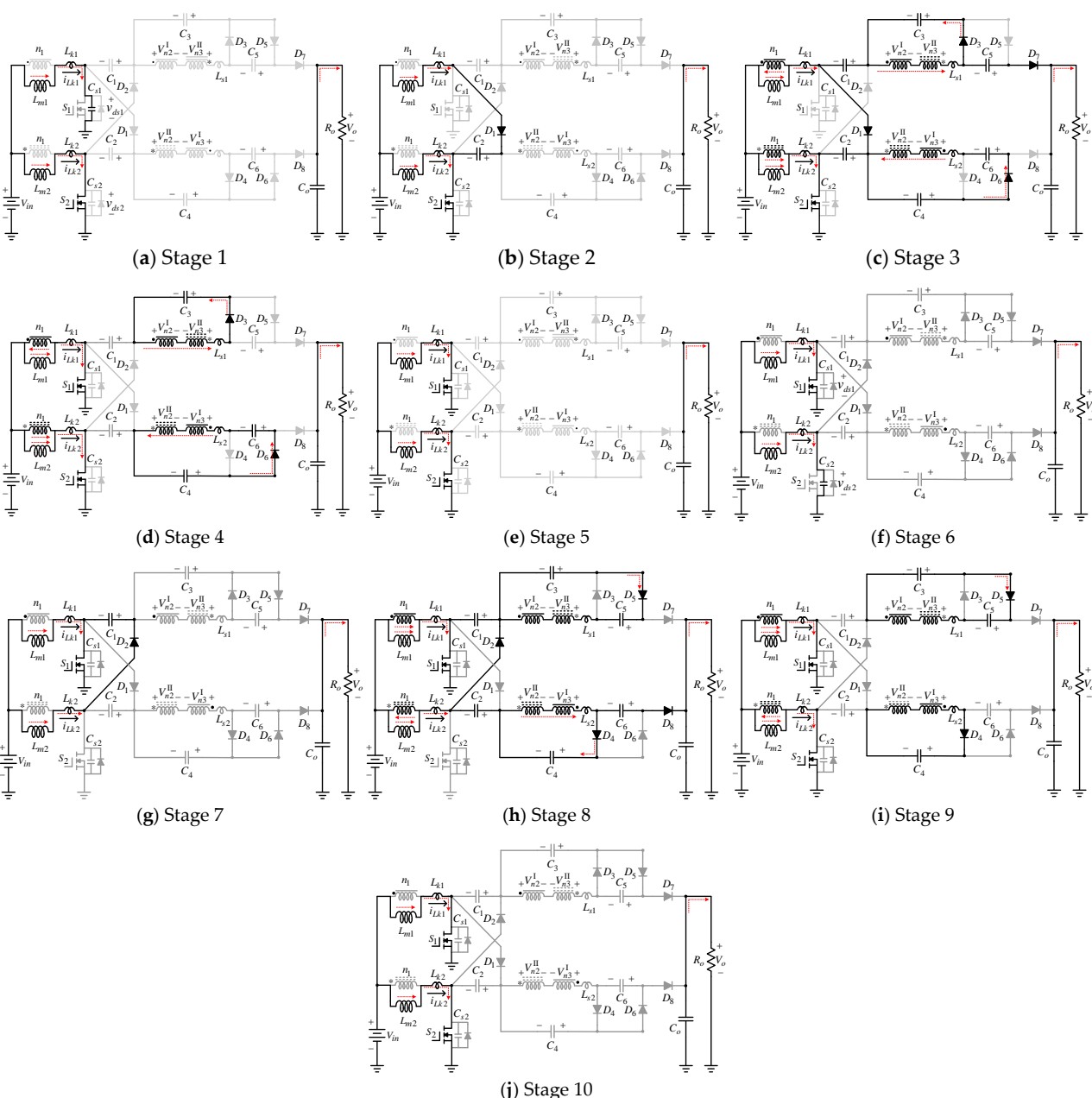

(**a**) Stage 1     (**b**) Stage 2     (**c**) Stage 3

(**d**) Stage 4     (**e**) Stage 5     (**f**) Stage 6

(**g**) Stage 7     (**h**) Stage 8     (**i**) Stage 9

(**j**) Stage 10

**Figure 5.** Operational stages of the proposed converter.

Stage 1 [$t_0 - t_1$]: As shown in Figure 5a, the switch $S_1$ is turned OFF at $t = t_0$, while the switch $S_2$ remains in the ON-state and all the diodes are in the OFF-state. The current of leakage inductance $L_{k1}$ starts to charge the parasitic capacitance $C_{S1}$ of switch $S_1$. The drain–source voltage $v_{ds1}$ of the switch $S_1$ increases from zero linearly due to the very small value of $C_{S1}$. The voltage relationship is

$$v_{ds1}(t) = v_{Cs1}(t_0) + \frac{1}{C_{s1}} \int_{t_0}^{t} i_{Cs1}(t) \, dt \cong \frac{i_{Lk1}(t_0)}{C_{s1}}(t - t_0) \tag{1}$$

Stage 2 [$t_1 - t_2$]: As shown in Figure 5b, the switch $S_1$ is in the OFF-state and the switch $S_2$ is in the ON-state. When the drain–source voltage $v_{ds1}$ increases to reach the voltage

of the clamp capacitor $C_2$ at $t = t_1$, the clamp diode $D_1$ starts to conduct and the voltage stress on the switch $S_1$ is clamped at the voltage $V_{C2}$. The clamp capacitor $C_2$ is charged by the leakage current $i_{Lk1}$. The reverse-biased voltage of the output diode $D_7$ decreases. The clamp capacitor voltage is

$$V_{C2}(t) = V_{C2}(t_1) + \frac{1}{C_2}\int_{t_1}^{t} i_{C2}(t)\, dt \cong V_{C2}(t_1) + \frac{1}{C_2}\int_{t_1}^{t} i_{Lk1}(t)\, dt \tag{2}$$

Stage 3 [$t_2 - t_3$]: As shown in Figure 5c, the switch $S_1$ remains in the OFF-state and $S_2$ remains in the ON-state. The reverse-biased voltage of the output diode $D_7$ decreases to zero and it begins to conduct at $t = t_2$. The current rising rate of the diode $D_7$ is controlled by the leakage inductances $L_{s1}$ and $L_{k1}$. As the current through the diode $D_7$ increases, the current through the diode $D_1$ decreases. In this stage, the clamp capacitor $C_1$, the secondary winding of WCCI 1, and the tertiary winding of WCCI 2 as well as the voltage-doubler capacitor $C_5$ play as voltage sources, which are in series to enlarge the output voltage. Part of the leakage current $i_{Lk1}$ flows to charge clamp capacitor $C_2$ through diode $D_1$ and switch $S_2$. Part of the leakage current $i_{Lk1}$ delivers to the output side through the clamp capacitor $C_1$, the windings of WCCI 1 and WCCI 2, voltage-doubler capacitor $C_5$, and output diode $D_7$. The input voltage source, coupled inductors, and capacitors $C_1$ and $C_5$ are in series to transfer energy to the output load. Moreover, the switched capacitor $C_4$ is discharged to voltage-doubler capacitor $C_6$ through diode $D_6$ and the windings of WCCI 1 and WCCI 2 in the second phase. The current relationships are given by

$$i_{Lm1} = i_{Lk1} + i_{n1} \tag{3}$$

$$i_{Lk1} = i_{D1} + i_{C1} = i_{D1} + i_{D7} \tag{4}$$

$$i_{n1} = \frac{n_2}{n_1}i_{D3} + \frac{n_3}{n_1}i_{D6} \tag{5}$$

$$i_{S2} = i_{Lk2} + i_{D1} \tag{6}$$

The energy stored in the magnetizing inductance $L_{m1}$ transfers to switched capacitor $C_3$ through the WCCIs in its phase and to voltage-doubler capacitor $C_6$ through the WCCIs in another phase.

Stage 4 [$t_3 - t_4$]: As shown in Figure 5d, the switch $S_1$ is turned on at $t = t_3$ and the switch $S_2$ remains in ON-state, while diodes $D_1$, $D_2$, $D_4$, $D_5$, $D_7$, and $D_8$ are reverse-biased. The current through leakage inductance $L_{k1}$ increases very quickly. The energy stored in magnetizing inductance $L_{m1}$ still transfers to the voltage multiplier cells when the condition $i_{Lk1} < i_{Lm1}$ is satisfied. Currents $i_{D3}$ and $i_{D6}$ decrease and their current falling rates are controlled by the leakage inductances.

Stage 5 [$t_4 - t_5$]: As shown in Figure 5e, switches $S_1$ and $S_2$ are both in the ON-state. When the leakage current $i_{Lk1}$ increases and reaches $i_{Lm1}$ at $t = t_5$, i.e., $i_{Lk1} = i_{Lm1}$, the energy transfer of magnetizing inductance ends. All the diodes are reverse-biased. The magnetizing inductances $L_{m1}$ and $L_{m2}$ as well as the leakage inductances $L_{k1}$ and $L_{k2}$ are linearly charged by the input voltage. The current relationships are given by

$$i_{Lk1}(t) = i_{Lk1}(t_4) + \frac{V_{in}}{L_{m1} + L_{k1}}(t - t_4) \tag{7}$$

$$i_{Lk2}(t) = i_{Lk2}(t_4) + \frac{V_{in}}{L_{m2} + L_{k2}}(t - t_4) \tag{8}$$

This stage ends when the switch $S_2$ is turned OFF at $t = t_5$. Due to the symmetrical structure, a similar operation proceeds in the next five stages.

## 3. Steady-State Analysis

To simplify the steady-state analysis of the proposed converter, the switches and diodes are assumed to be ideal. The leakage inductances are neglected. All capacitors

are large enough, so the voltages on the capacitors are considered to be constant in one switching period. Due to the symmetrical structure of the converter circuit, it is feasible to consider the values of relevant components to be equal such as $L_{m1} = L_{m2}$, $C_1 = C_2$, $C_3 = C_4$, and $C_5 = C_6$. Only stages 3, 5, 8, and 10 are considered in the steady-state analysis because the time transitions of stages 1, 2, 4, 6, 7, and 9 are significantly short.

### 3.1. Voltage Gain Derivation

Applying the volt-second balance principle to the magnetizing inductances $L_{m1}$ and $L_{m2}$, the voltages on the clamp capacitors $C_1$ and $C_2$ can be derived from

$$V_{C1} = V_{C2} = \frac{1}{1-D}V_{in} \tag{9}$$

where $D$ is the duty ratio. The voltages on the switched capacitors and the voltage-doubler capacitors can be derived from the KVLs around the loops of the equivalent circuits of Stage 3 and Stage 8, respectively.

$$V_{C3} = V_{n3}^{II} - V_{n2}^{I} = nV_{in} - n(V_{in} - V_{C2}) = \frac{n}{1-D}V_{in} \tag{10}$$

$$V_{C4} = V_{n3}^{I} - V_{n2}^{II} = nV_{in} - n(V_{in} - V_{C1}) = \frac{n}{1-D}V_{in} \tag{11}$$

$$V_{C6} = V_{C4} + V_{n2}^{II} - V_{n3}^{I} = \frac{2n}{1-D}V_{in} \tag{12}$$

$$V_{C5} = V_{C3} + V_{n2}^{I} - V_{n3}^{II} = \frac{2n}{1-D}V_{in} \tag{13}$$

The output voltage can be derived from the KVL around the loop of the equivalent circuit of Stage 3 and Equations (9)–(13)

$$V_{o} = V_{C5} + V_{C3} + V_{C1} + V_{C2} = \frac{3n+2}{1-D}V_{in} \tag{14}$$

Consequently, the voltage gain is given as below

$$\frac{V_o}{V_{in}} = \frac{3n+2}{1-D} \tag{15}$$

It is clear that there are two degrees of freedom to design the voltage gain: duty ratio and turns ratio of the winding-cross-coupled inductor (WCCI). The relation curves of voltage gain versus the duty ratio and the turns ratio of WCCI are shown in Figure 6. If the duty ratio is 0.6, the voltage gain is 12.5 with turns ratio $n = 1$. Therefore, the proposed converter can achieve high voltage gain with a proper duty ratio.

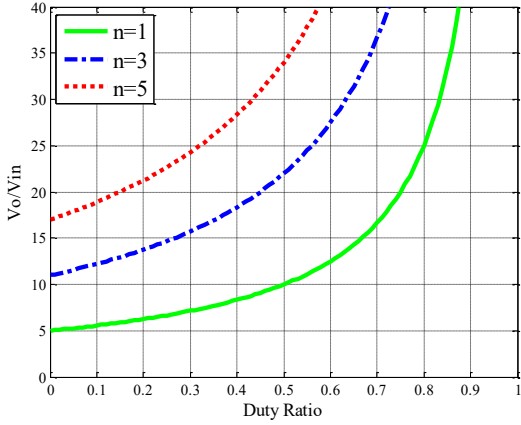

**Figure 6.** Relation curves of voltage gain versus duty ratio and turns ratio.

### 3.2. Voltage Stresses on Semiconductors

Based on the operation principles and the results of Equations (9)–(13), the voltage stresses on the switches and the diodes can be derived as

$$V_{S1} = V_{S2} = \frac{1}{1-D}V_{in} = \frac{1}{3n+2}V_o \tag{16}$$

$$V_{D1} = V_{D2} = \frac{2}{1-D}V_{in} = \frac{2}{3n+2}V_o \tag{17}$$

$$V_{D3} = V_{D4} = V_{D5} = V_{D6} = \frac{2n}{1-D}V_{in} = \frac{2n}{3n+2}V_o \tag{18}$$

$$V_{D7} = V_{D8} = \frac{2n+1}{1-D}V_{in} = \frac{2n+1}{3n+2}V_o \tag{19}$$

It can be seen from Equations (16)–(19) that the voltage stresses on the switches and diodes are determined by the turn ratio of the WCCIs and the output voltage. The switch voltage stress decreases as the turn ratio $n$ increases. The switch voltage stress is only one-fifth of the output voltage with $n = 1$. The low-voltage-rated MOSFETs with low $R_{ds(on)}$ can be adopted to reduce the conduction losses compared with the conventional boost converter. The relation curves between the normalized voltage stress ratio and the turns ratio are shown in Figure 7. It can be seen that the voltage stress ratio of diodes $D_1$ and $D_2$ decreases with the increase in the turns ratio. The voltage stress ratio of diodes $D_3 \sim D_8$ increases with the increase in turns ratio. Their maximum voltage stress ratio approaches 0.67. Therefore, the diode voltage stress always remains lower than the output voltage. The low-voltage-rated diodes with low forward voltage drop can be adopted to reduce the conduction losses.

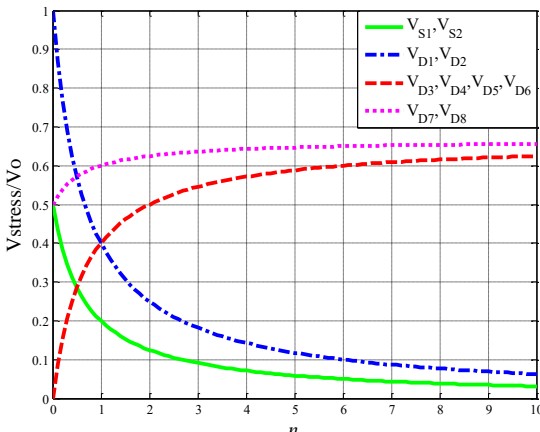

**Figure 7.** Relation curves of the normalized power-device voltage stress ratio versus the turns ratio.

### 3.3. Performance Comparison

The performance comparison between the proposed converter and the existing converters [21–24] is shown in Table 1. The voltage gain of the proposed converter is the highest and the voltage stress on the switches is the lowest. The highest voltage stress on the diodes of the proposed converter is lower than that of the converters in [21,22,24].

**Table 1.** Converter performance comparison.

| High Voltage Gain Converter | Converter in [21] | Converter in [22] | Converter in [23] | Converter in [24] | Proposed Converter |
|---|---|---|---|---|---|
| Voltage gain | $\frac{2n+2}{1-D}$ | $\frac{3n+1}{1-D}$ | $\frac{2n+2}{1-D}$ | $\frac{2n+2}{1-D}$ | $\frac{3n+2}{1-D}$ |
| Voltage stress | $\frac{V_o}{2n+2}$ | $\frac{V_o}{3n+1}$ | $\frac{V_o}{2n+2}$ | $\frac{V_o}{2n+2}$ | $\frac{V_o}{3n+2}$ |
| Maximum diode voltage stress | $\frac{(2n+1)V_o}{2n+2}$ | $\frac{2nV_o}{3n+1}$ | $\frac{(2n+1)V_o}{2n+2}$ | $\frac{(2n+1)V_o}{2n+2}$ | $\frac{(2n+1)V_o}{3n+2}$ |

**Table 1.** *Cont.*

| High Voltage Gain Converter | Converter in [21] | Converter in [22] | Converter in [23] | Converter in [24] | Proposed Converter |
|---|---|---|---|---|---|
| Number of switches | 2 | 2 | 2 | 2 | 2 |
| Number of diodes | 6 | 8 | 6 | 6 | 8 |
| Number of capacitors | 5 | 7 | 5 | 5 | 7 |
| Number of coupled inductor | 2 | 2 | 2 | 2 | 2 |
| Voltage gain $n = 1,\ D = 0.6$ | 10 | 10 | 10 | 10 | 12.5 |

## 4. Converter Design Guidelines

### 4.1. WCCIs Turns Ratio Design

The turns ratio design is important because it determines the voltage gain of the proposed converter and the voltage stresses of semiconductors. An appropriate turns ratio can be designed according to the Equation (15) if a proper duty ratio is selected, which is given by

$$n = \frac{(1 - D)V_o}{3V_{in}} - \frac{2}{3} \tag{20}$$

Once the turns ratio is designed, the voltage stresses on the switches and the diodes can be determined from the Equations (16)–(19).

### 4.2. Magnetizing Inductance Design

The magnetizing inductance of the coupled inductor is designed to operate in CCM and ripple current consideration. Let $I_{Lm}$ denote the average current through the magnetizing inductor and $\Delta i_{Lm}$ denotes its ripple current, then the condition of CCM operation is given by

$$I_{Lm} - \frac{1}{2}\Delta i_{Lm} > 0 \tag{21}$$

Based on the operation principles, the magnetizing inductances of WCCIs can be calculated as

$$L_m > \frac{D(1 - D)^2 R_o}{(3n + 1)^2 f_s} \tag{22}$$

where $f_s$ is the switching frequency.

### 4.3. Capacitor Design

The capacitances are designed to suppress the voltage ripple to an acceptable level. The rated voltage of each capacitor can be obtained from Equations (9)–(13). Once the ripple voltage ratio $\Delta V_C / V_C$ is determined, the selections of the corresponding capacitors are given by

$$C_1 = \frac{3n + 2}{2R_o f_s (\Delta V_{C1}/V_{C1})}, \quad C_2 = \frac{3n + 2}{2R_o f_s (\Delta V_{C2}/V_{C2})} \tag{23}$$

$$C_3 = \frac{3n + 2}{2nR_o f_s (\Delta V_{C3}/V_{C3})}, \quad C_4 = \frac{3n + 2}{2nR_o f_s (\Delta V_{C4}/V_{C4})} \tag{24}$$

$$C_5 = \frac{3n + 2}{4nR_o f_s (\Delta V_{C5}/V_{C5})}, \quad C_6 = \frac{3n + 2}{4nR_o f_s (\Delta V_{C6}/V_{C6})} \tag{25}$$

The output capacitor selection is obtained as

$$C_o = \frac{2D - 1}{R_o f_s (\Delta V_{Co}/V_{Co})} \tag{26}$$

## 5. Closed-Loop Controller Design

In order to diminish the effect of the variations in input voltage and load on the output voltage, there are some reported controllers used in the closed-loop controlled system design, such as sliding-mode control [25,26], model predictive control [27], and voltage dual-loop control [28]. However, the implementation of these control methods is more complicated. In this paper, a voltage-mode control method is used and designed for the closed-loop controlled system to keep a regulated output voltage in spite of the variations in the input voltage and output load. The control method is popular and low cost by using the pulse-width modulated controller integrated circuit (PWM IC). The block diagram of the closed-loop control system is shown in Figure 8, where $C(s)$ is the controller transfer function; $1/V_P$ is the pulse-width modulator gain; $V_P$ is the amplitude of sawtooth waveform in the PWM circuit; $P(s)$ is the duty ratio-to-output transfer function of the proposed converter; and $K$ is the feedback gain of sensing output voltage.

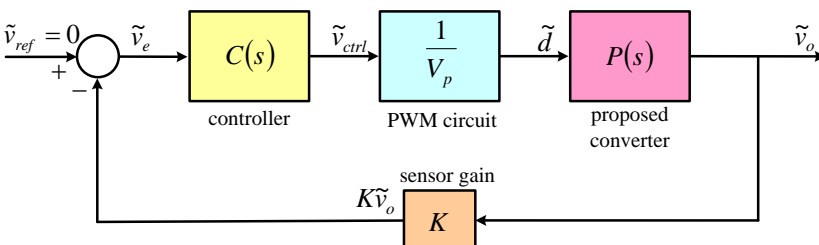

**Figure 8.** Diagram of the feedback control system.

The frequency response analyzer NF FRA51602 is employed to measure the Bode plot from the control signal $\tilde{v}_{ctrl}$ to the sensing output voltage signal $K\tilde{v}_o$ at the operating point of the converter prototype. The measured Bode plot is shown in Figure 9 in a red line. Then, the curve-fitting method by the MATLAB R2021a software is used to establish the control-to-sensing output transfer function of $G(s)$, where

$$G(s) = \frac{K\tilde{v}_o(s)}{\tilde{v}_{ctrl}(s)} = \frac{K}{V_p}P(s) \tag{27}$$

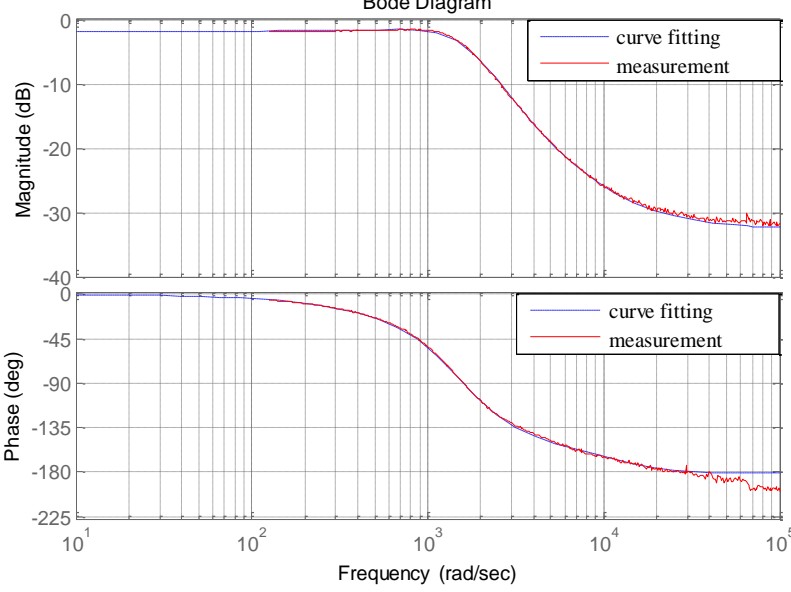

**Figure 9.** Comparison of frequency responses.

The small-signal transfer function by the curve-fitting method is obtained as

$$G(s) = \frac{0.023572 \times (s + 6000)(s + 24000)(9400 - s)}{(s + 17050)(s^2 + 1945.6s + 2310400)} \tag{28}$$

The Bode plot of the measured result (in red line) together with the Bode plot of the transfer function $G(s)$ (in blue line) are shown in Figure 9. It can be seen that good agreement of the curves up to the frequency $3 \times 10^4$ rad/sec has been obtained. Therefore, the transfer function $G(s)$ is feasible to be used in the controller design.

A Type III controller is designed based on the *K* factor approach [29] in this article, which is widely used in the control loop for the power converter. Especially, it is employed for the controlled plant that has a big phase lag around the gain crossover frequency. The controller circuit, known as the Type III amplifier, is shown in Figure 10 and its small-signal transfer function can be written in the following form.

$$\frac{\widetilde{v}_{ctrl}(s)}{K\widetilde{v}_o(s)} = -\frac{R_1 + R_3}{R_1 R_3 C_2} \frac{\left(s + \frac{1}{R_2 C_1}\right)\left(s + \frac{1}{(R_1 + R_3)C_3}\right)}{s\left(s + \frac{1}{R_2 C_1 C_2/(C_1 + C_2)}\right)\left(s + \frac{1}{R_3 C_3}\right)} \tag{29}$$

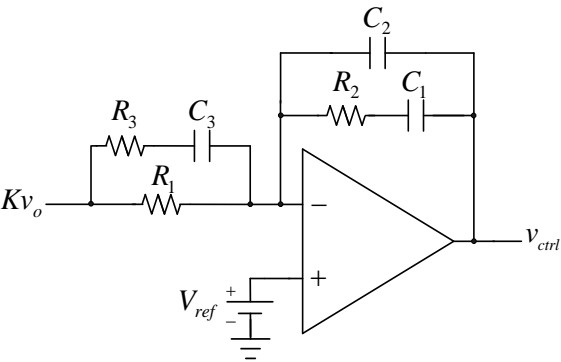

**Figure 10.** Type III amplifier.

Assuming $C_2 \ll C_1$ and $R_3 \ll R_1$, then

$$\frac{\widetilde{v}_{ctrl}(s)}{K\widetilde{v}_o(s)} \approx -\frac{1}{R_3 C_2} \frac{\left(s + \frac{1}{R_2 C_1}\right)\left(s + \frac{1}{R_1 C_3}\right)}{s\left(s + \frac{1}{R_2 C_2}\right)\left(s + \frac{1}{R_3 C_3}\right)} \tag{30}$$

The controller has three poles and two zeros, including one pole at the origin. From a viewpoint, it consists of an integrator and two sets of phase leaders. The integrator is helpful to achieve zero steady-state errors for the constant reference input. The phase leader can provide the required phase boost to maintain a reasonable phase margin and to make the control system stable. In order to meet the specifications of a gain crossover frequency of 1 kHz and phase margin of more than $50°$, the controller is designed as

$$C(s) = 350000 \times \frac{(s + 1399)(s + 1361)}{s(s + 27040)(s + 28570)} \tag{31}$$

The six passive components of the Type III amplifier are implemented with $R_1 = 100$ kΩ, $R_2 = 330$ kΩ, $R_3 = 7.2$ kΩ, $C_1 = 1.8$ nF, $C_2 = 0.13$ nF, and $C_3 = 5.9$ nF.

The Bode plots of the open-loop transfer function $T_{OL}(s) = C(s)G(s)$, the controller $C(s)$, and the plant $G(s)$ are shown in Figure 11. It can be seen that the control system has a 1 kHz gain crossover frequency and a $57°$ phase margin. The controller provides a maximum phase boost at the crossover frequency.

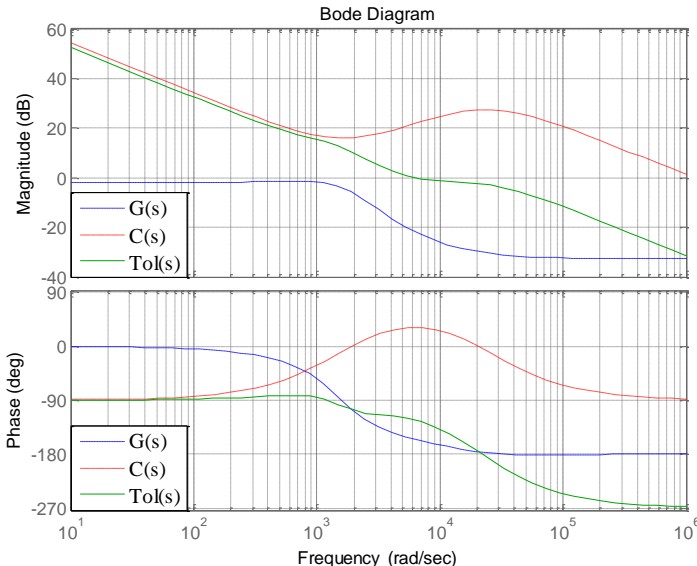

**Figure 11.** Bode plots of the transfer functions $T_{OL}(s)$, $C(s)$, and $G(s)$.

## 6. Experimental Results

A 1000 W laboratory prototype with an input voltage of 36 V and an output voltage of 400 V is implemented for performance verification. The PWM IC TL494 is used in the prototype, which is low-cost and easy to compensate. The reliability of the control system is improved. The parameters of the converter prototype are shown in Table 2. The primary, secondary, and tertiary winding are made of 20 turns. The powder core CH467125 is used in the winding-cross-coupled inductors. The following experimental waveforms shown in Figures 12–15 are measured at a full-load condition.

The experimental waveforms of the gate signals $v_{gs1}$ and $v_{gs2}$ and the drain-to-source voltages $v_{ds1}$ and $v_{ds2}$ of the switches are shown in Figure 12a. It is verified that the converter achieves high voltage gain over 11 times without an extreme duty ratio. The switch voltage stress is about 80 V, which is only one-fifth of the output voltage and agrees with the analysis results of Equation (16). Consequently, the low-voltage-rated MOSFETs with low $R_{ds(on)}$ can be adopted to reduce the conduction losses. The experimental waveforms of the input current $i_{in}$ and the leakage currents $i_{Lk1}$ and $i_{Lk2}$ are shown in Figure 12b. The average currents of $i_{Lk1}$ and $i_{Lk2}$ are almost equal with the help of winding-cross-coupled inductors and converter configuration. The input current is equally shared in two phases such that the device current stresses are reduced. Furthermore, the leakage current ripples are 20.53 A and the input current ripple is only 3.6 A. The input current ripple is greatly reduced owing to the interleaved operation.

**Table 2.** Parameters of the converter prototype.

| Components | Parameters |
|---|---|
| Switching frequency $f_s$ | 40 kHz |
| Turns ratio of coupled inductor $n$ | 1 |
| Magnetizing inductances $L_{m1}$, $L_{m2}$ | 140 µH |
| Leakage inductances $L_{k1}$, $L_{k2}$ | 0.6 µH |
| Clamp capacitors $C_1$, $C_2$ | 22 µF |
| Switched capacitors $C_3$, $C_4$ | 22 µF |
| Voltage-doubler capacitors $C_5$, $C_6$ | 22 µF |
| Output capacitor $C_o$ | 32 µF |
| Switches $S_1$, $S_2$ | IRFP4227 |
| Diodes $D_1$, $D_2$, $D_3$, $D_4$, $D_5$, $D_6$ | MBR20200CT |
| Diodes $D_7$, $D_8$ | STTH3003CW |

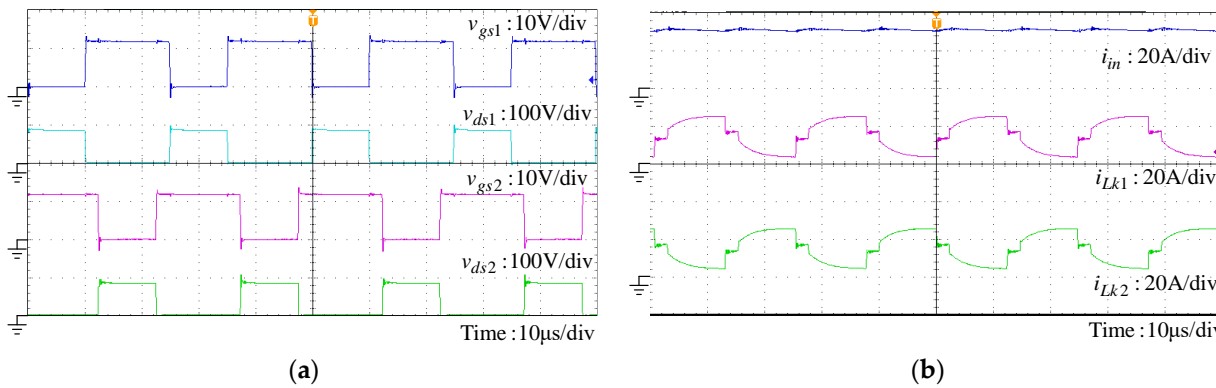

**Figure 12.** Experimental waveforms: (**a**) $v_{gs1}$, $v_{ds1}$, $v_{gs2}$, and $v_{ds2}$; (**b**) $i_{in}$, $i_{Lk1}$, and $i_{Lk2}$.

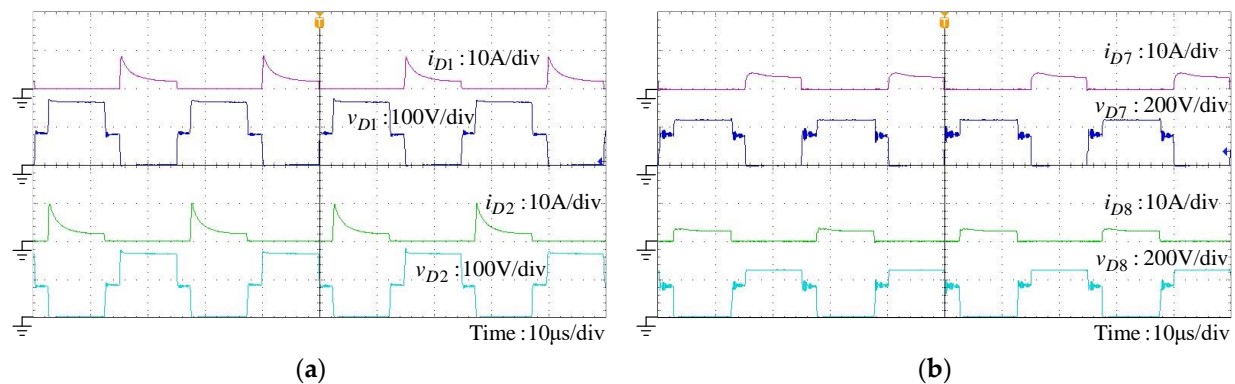

**Figure 13.** Experimental waveforms: (**a**) $i_{D1}$, $v_{D1}$, $i_{D2}$, and $v_{D2}$; (**b**) $i_{D7}$, $v_{D7}$, $i_{D8}$, and $v_{D8}$.

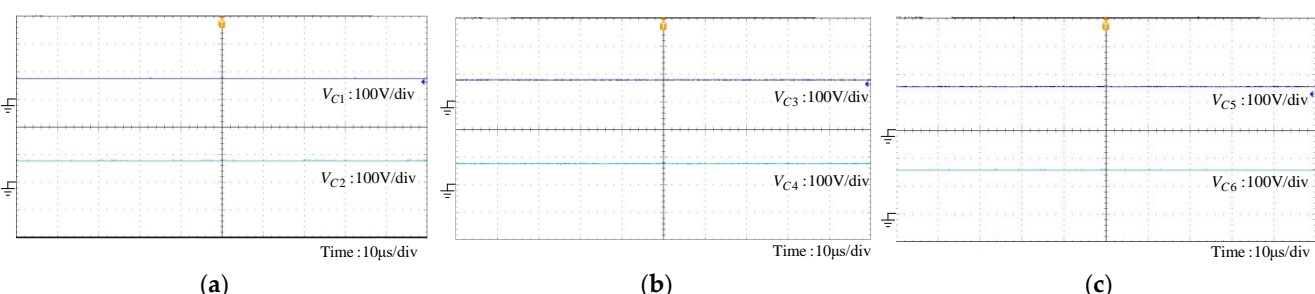

**Figure 14.** Experimental waveforms: (**a**) $V_{C1}$ and $V_{C2}$; (**b**) $V_{C3}$ and $V_{C4}$; (**c**) $V_{C5}$ and $V_{C6}$.

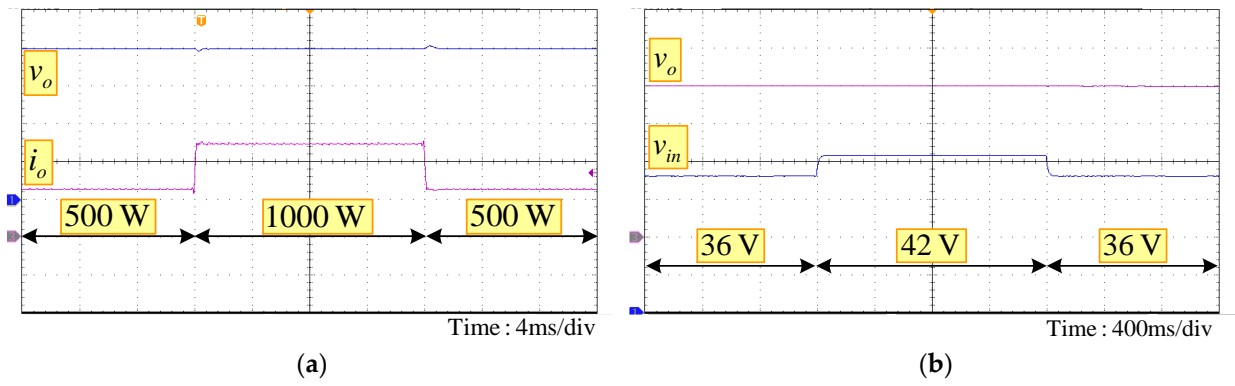

**Figure 15.** Output voltage response. (**a**) Step load variations; (**b**) Input voltage variations.

The experimental waveforms of the voltage and current on the clamp diodes and the output diodes are demonstrated in Figure 13a,b. The reverse recovery problem of each

diode is alleviated due to the existence of leakage inductances. Furthermore, the voltage stress on the clamp diodes $D_1$ and $D_2$ is about 160 V and that on the output diodes $D_7$ and $D_8$ is 240 V, which is much lower than the output voltage and consistent with the analysis results of Equations (17) and (19).

The voltage waveforms of the clamp capacitors $C_1$ and $C_2$, the switched capacitors $C_3$ and $C_4$, and the voltage-doubler capacitors $C_5$ and $C_6$, are shown in Figure 14. The voltages $V_{C1}$, $V_{C2}$, $V_{C3}$, and $V_{C4}$ are about 80 V and the voltages $V_{C5}$ and $V_{C6}$ are about 160 V, which are consistent with the analysis results of Equations (9)–(13).

The experimental waveforms of the output voltage and the output current under the step load variation between 500 W and 1000 W are illustrated in Figure 15a. Furthermore, the experimental waveforms of the output voltage and the input voltage variation between 36 V and 42 V are shown in Figure 15b. It can be seen that the transient voltage ripple of the output voltage is very small. The output voltage regulation performance is excellent because the controller in the closed-loop control system is well-designed.

The experimental conversion efficiency at different loads is measured by the power analyzer HIOKI 3390 (HIOKI E.E. Corporation, Nagano, Japan), as shown in Figure 16. The maximum efficiency is 97.40% at 200 W. The efficiency of 90.40% is achieved at a 1000 W full-load. The results show that the proposed converter has an efficiency higher than 90% for the overall load conditions.

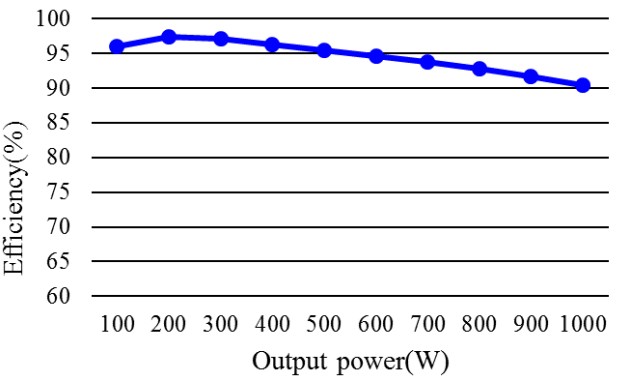

**Figure 16.** Measured efficiency at different loads.

It is worth mentioning that the thermal management of power converters has gained significant attention due to high power density and reliability considerations [30,31]. Cooling technologies have been a research area in the power electronic converter [32]. Therefore, thermal management is a topic worthy of further research in the future.

## 7. Conclusions

A new interleaved high voltage gain DC-DC converter with winding-cross-coupled inductors and voltage multiplier cells is proposed for photovoltaic systems in this article. The operation principles, steady-state analysis, closed-loop controller design, and experimental verifications of the proposed converter are presented in detail. The high voltage gain can be achieved for the proposed converter with a proper duty ratio operation. The switch voltage stress is low such that the low-voltage-rated MOSFETs with low on-resistance can be adopted to reduce the conduction losses. Moreover, the diode voltage stress is low such that the diodes with low forward voltage drop can be adopted to reduce the conduction losses. The interleaved operation reduces the input current ripple. The clamp circuit can clamp the switch voltage stress and recycle the leakage energy such that the switch turned-off voltage spike can be avoided. The winding-cross-coupled inductor is helpful in making the current auto-balance of two phases. In addition, a feedback controller is designed to diminish the effect of the input voltage and load variations on the output voltage. Finally, a 1000 W converter prototype is implemented and the experimental results are given to validate the converter performance and the theoretical analysis. The proposed converter



can clearly meet the requirements of high voltage gain and high-efficiency conversion of photovoltaic systems.

**Author Contributions:** This paper is a collaborative work of all authors. Conceptualization, S.-J.C. and C.-M.H.; methodology, S.-J.C. and S.-P.Y.; software, S.-D.L. and C.-H.C.; Investigation, S.-J.C. and S.-P.Y.; validation, S.-J.C. and S.-D.L.; Writing—original draft preparation, S.-J.C. and S.-D.L.; Writing—review and editing, S.-J.C. and C.-H.C.; Supervision, C.-M.H. All authors have read and agreed to the published version of the manuscript.

**Funding:** This research is funded by the National Science and Technology Council, Taiwan, under grant No. NSTC 112-2221-E-168-010.

**Data Availability Statement:** Data are contained within the article.

**Conflicts of Interest:** The authors declare no conflicts of interest.

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
