# Peer review of "Interleaved High Voltage Gain DC-DC Converter with Winding-Cross-Coupled Inductors and Voltage Multiplier Cells for Photovoltaic Systems"

_electronics, doi:10.3390/electronics13101851_

Round 1

Reviewer 1 Report

Comments and Suggestions for Authors

In this paper, an interleaved high voltage gain DC-DC converter with winding-cross-coupled inductors (WCCIs) and voltage multiplier cells is proposed for photovoltaic systems.The closed-loop controlled system of the proposed con-verter is designed to diminish the effect of the variations of input voltage and load on the output 20 voltage. The control methods should be discussed more clearly especially with existed publications such as DOI: 10.1109/TASE.2024.3369905 and DOI: 10.1109/TASE.2023.3334435.

Comments on the Quality of English Language

More relevant english expression should be corrected.

Reviewer 2 Report

Comments and Suggestions for Authors

Article presents the interleaved high voltage gain DC-DC converter with winding cross coupled inductors and voltage multiplier cells. The operation principles along with steady-state analysis, closed loop control and experimental results. The experimental waveforms are presented and supported with theoretical design procedures. The 1000W converter prototype is implemented and experimental results are given to validate the performance of the converter.

Other few comments and suggestions

-       On page 1, Text: The converter achieves high voltage gain without operating at extreme duty ratio. (suggest to rewrite the sentence as the duty ratio is in range from 0-1. Nothing extreme about that. The operation of the converter at high duty ratios might be extreme)

-       On page 14, in Figure 16 a, and b the time scale is missing.

-       On page 7, Rewrite the text: The capacitor voltages are considered to be constant in one switching period because of their large enough capacitances.

-       On page 7,  Rewrite the text: The time durations of stages 1, 2, 4, 6, 7, 9 are significantly short and they are ignored in the steady-state analysis.

Comments on the Quality of English Language

Minor editing of English language required.

Reviewer 3 Report

Comments and Suggestions for Authors

This article presents a novel DC-DC converter designed for photovoltaic systems using winding-cross-coupled inductors and voltage multiplier cells.

By optimizing the duty cycle, the converter achieves significant voltage gain while minimizing switch voltage stress, allowing the use of low-voltage MOSFETs with reduced conduction losses. In addition, diode voltage stress is minimized, allowing the use of diodes with lower forward voltage drop.

A 1000 W prototype validates the converter's performance and theoretical analysis, demonstrating its suitability for high voltage gain and efficient conversion in photovoltaic systems.

The topic of the article is relevant. I have no comments on the manuscript. The methods used are adequately described. Guidelines for designing the converter are presented. The conclusions support the results.

Reviewer 4 Report

Comments and Suggestions for Authors

The paper proposed an innovative design for a high voltage gain DC-DC converter suitable for photovoltaic systems. It meets the need for high voltage gain without operating at extreme duty ratios, which is a significant improvement over conventional converters. However, I think the paper does not meet the appropriate quality standards to be accepted. Here are some comments.

1)While the control system is designed to regulate the output voltage, its impact on the overall system cost and reliability could be further discussed.

2)It would be better if the author can provide a more detailed discussion on how the proposed converter outperforms existing solutions in terms of specific applications, especially under the background of photovoltaic systems.

3)The paper does not mention the thermal management of the converter, which is crucial for high power applications. 

Round 2

Reviewer 4 Report

Comments and Suggestions for Authors

The revision is good